# Strategic Scaling of Test-Time Compute: A Bandit Learning Approach

**Bowen Zuo**   **Yinglun Zhu**[†]
University of California, Riverside
{bzuo002, yzhu}@ucr.edu

## Abstract

Scaling test-time compute has emerged as an effective strategy for improving the performance of large language models. However, existing methods typically allocate compute uniformly across all queries, overlooking variation in query difficulty. To address this inefficiency, we formulate test-time compute allocation as a novel bandit learning problem and propose adaptive algorithms that estimate query difficulty on the fly and allocate compute accordingly. Compared to uniform allocation, our algorithms allocate more compute to challenging queries while maintaining accuracy on easier ones. Among challenging queries, our algorithms further learn to prioritize solvable instances, effectively reducing excessive computing on unsolvable queries. We theoretically prove that our algorithms achieve better compute efficiency than uniform allocation and empirically validate their effectiveness on math and code benchmarks. Specifically, our algorithms achieve up to an 11.10% performance improvement (15.04% relative) on the MATH-500 dataset, up to 10.82% (14.44% relative) on the AIME25 dataset, and up to an 11.23% performance improvement (15.29% relative) on the LiveCodeBench dataset.

## 1 Introduction

Recent advances in large language models (LLMs) have shifted attention from training-time compute (Kaplan et al., 2020; Hoffmann et al., 2022; Chowdhery et al., 2022) to test-time compute (Wei et al., 2023; Yao et al., 2023; Madaan et al., 2023; Agarwal et al., 2024; Muennighoff et al., 2025) as a means of improving model performance. Test-time scaling methods such as Best-of-$N$ sampling (Brown et al., 2024; Snell et al., 2024) and consistency checking (Wang et al., 2022) enhance output quality by generating multiple responses and selecting the most promising one. This selection process can be strengthened using high-quality reward oracles (Cobbe et al., 2021; Uesato et al., 2022; Lightman et al., 2023; Zhang et al., 2025a). These methods have achieved strong empirical gains without additional model training. For instance, as noted in OpenAI's o1 release report (OpenAI, 2024), repeated sampling with 64 generations improves accuracy on the 2024 AIME competition math dataset from 74.4% to 83.3%—a nearly 9% gain without any model updates.

Despite recent advances, most test-time scaling techniques still allocate compute *uniformly across all queries* (Brown et al., 2024; Snell et al., 2024), ignoring the inherent variability in query difficulty. This one-size-fits-all strategy is inefficient: simple arithmetic questions receive the same compute as multi-step reasoning tasks, leading to wasted resources on easy queries and insufficient budget on hard ones. Ideally, one should allocate *just enough compute* to confidently solve easy queries and *reallocate the remaining budget* to harder ones. While recent work has begun exploring adaptive test-time strategies, existing methods either (1) focus on compute allocation *within a single query* (Sun et al., 2024; Manvi et al., 2024; Tan et al., 2025), or (2) rely on *two-stage procedures* (Damani et al., 2024; Wang et al., 2025b) that require training an auxiliary model (or pre-compute allocation) in the first stage to guide later allocation decisions.

In this work, we introduce a new perspective—*strategic scaling of test-time compute*—in which compute is adaptively allocated *across* a set of queries based on their estimated difficulty. We formulate test time compute allocation as a *fully adaptive* pure-exploration-style bandit learning

---

[†]Project lead and corresponding author.

Figure 1: Comparison between our algorithm and baselines. *First:* Accuracy comparison on MATH-500 with `Llama-3.1-8B-Instruct`. *Second:* Accuracy comparison on MATH-500 with `Gemini-2.5-flash-lite`. *Third:* Accuracy comparison on AIME25 with `Qwen3-4B`. *Fourth:* Coverage comparison on LiveCodeBench with `DeepSeek-R1-Distill-Llama-8B`.

problem (Bubeck et al., 2009; Jamieson & Nowak, 2014; Locatelli et al., 2016; Zhu et al., 2020), treating each query as an action and allocating compute sequentially to *maximize the number of queries answered correctly within a fixed budget*. Our adaptive algorithms estimate query difficulty on the fly and prioritize compute for those most likely to benefit from additional inference. Empirically, our method achieves up to 11.10% absolute (15.04% relative) improvement on the MATH-500 dataset (Lightman et al., 2023; Hendrycks et al., 2021), 10.82% absolute (14.44% relative) on AIME25 (AIME, 2025), and 11.23% absolute (15.29% relative) on LiveCodeBench (Jain et al., 2024)—all under the same compute budget as baselines. Fig. 1 provides a high-level comparison between our method and various baselines.

**Contributions.** We summarize our main contributions below:

1. We formulate LLM test-time compute allocation as a novel bandit learning problem, bridging test-time scaling and bandit learning communities. This formulation grounds strategic test-time scaling in a precise decision-theoretic framework.

2. We propose a general algorithmic framework for strategic compute allocation, supporting flexible exploration strategies—including a novel entropy-based rule. Our framework naturally extends to incorporate alternative aggregation methods and handles both streaming and token-constrained settings. We further provide theoretical insights into the efficiency gains of our adaptive approach.

3. We conduct extensive experiments on math and code benchmarks and show that our algorithms consistently outperform baselines. Further analyses demonstrate that our algorithms adaptively allocate compute to harder queries in standard settings, and to solvable queries in scenarios containing both solvable and unsolvable instances, effectively avoiding compute waste.

**Paper organization.** The rest of this paper is organized as follows. We introduce the problem of strategic test-time compute allocation in Section 2. Our solution is presented in Section 3, including the bandit formulation, algorithmic framework, extensions, and theoretical analysis. Empirical results are in Section 4, covering main results, analyses, and ablations. We conclude in Section 5. Related work, formal proofs, and additional experimental details and results are provided in the Appendix.

## 2 PROBLEM SETTING

Let $p$ denote a language model, which takes a query $x \in \mathcal{X}$ as input and generates a response $y \sim p(\cdot \mid x)$. Recent studies show that scaling up the test-time compute can significantly improve the performance of LLMs across a variety of tasks (Snell et al., 2024). In this context, we consider the amount of test-time compute as the total number of responses generated by the language model. For example, given query $x \in \mathcal{X}$ and a compute budget of $N$, the model can generate a set of $N$ responses $g(x; N) := \{y_1, \cdots, y_N\}$, where each response $y_i \sim p(\cdot \mid x)$ is sampled from the conditional distribution $p(\cdot \mid x)$. A *reward oracle* $r : \mathcal{X} \times \mathcal{Y} \to [0, 1]$ is used to evaluate the quality of each generation; the reward oracle can be instantiated by either a ground truth verifier or a learned reward model (Cobbe et al., 2021; Uesato et al., 2022; Lightman et al., 2023; Zhang et al., 2025a). When the evaluation metric requires a single response as the output, test-time compute methods such as the Best-of-$N$ algorithm (Brown et al., 2024) use the reward oracle to score each response and return the one with the highest score. Specifically, given a set of responses $g(x; N) = \{y_1, \cdots, y_N\}$ and

letting $r(x, y_i)$ denote the score of response $y_i$, the final output $f(x; N) := f(g(x; N))$ is defined as:

$$f(x; N) = y_{i^\star}, \quad \text{where} \quad i^\star := \arg\max_{i \in [N]} r(x, y_i).$$

While scaling test-time compute can improve performance, existing methods primarily focus on *uniform allocation of compute budget*. Specifically, given a set of queries $S = \{x_1, \cdots, x_n\}$ and total compute budget $B := n\bar{B}$, existing approaches assign the same compute budget $\bar{B}$ to each query $x_i$ and generate the final outputs $\{(x_1, f(x_1; \bar{B})), \cdots, (x_n, f(x_n; \bar{B}))\}$. This uniform allocation is inefficient: it ignores differences in query difficulty and *assigns the same compute to both easy and hard queries*.

## 2.1 Strategic test-time compute allocation

To address the limitations of uniform allocation, we study the problem of *strategic test-time compute allocation*—how to *adaptively* allocate a total compute budget across a set of queries to *maximize the fraction of correctly answered queries*. Let $B$ denote the total compute budget and $S = \{x_1, \cdots, x_n\}$ be a set of $n$ queries. Let $\mathsf{Metric} \in [0, 1]$ be an evaluation metric and $c(x_i)$ be the compute allocated to query $x_i$. The goal is to maximize the overall performance subject to a budget constraint:

$$\max_{\{c(x_i)\}_{i=1}^n} \frac{1}{n} \sum_{i=1}^n \mathsf{Metric}\big(x_i; c(x_i)\big) \quad \text{subject to} \quad \sum_{i=1}^n c(x_i) \le B. \tag{1}$$

We consider two popular evaluation metrics: coverage and accuracy. Given a compute allocation $c(x_i)$, *coverage* evaluates whether any of the $c(x_i)$ generations in $g(x_i; c(x_i))$ correctly answers the query $x_i$, while *accuracy* evaluates whether the final output $f(x_i; c(x_i))$ is correct. These metrics are defined as:

$$\mathsf{Coverage}(x_i; c(x_i)) := \mathbb{I}\{\text{there exists } y \in g(x_i; c(x_i)) \text{ that correctly answers query } x_i.\}$$
$$\mathsf{Accuracy}(x_i; c(x_i)) := \mathbb{I}\{f(x_i; c(x_i)) \text{ correctly answers query } x_i.\}$$

The key challenge in Eq. (1) is to adaptively allocate compute budget $c(x_i)$ to each query $x_i$ *under uncertainty*—that is, without knowing in advance the difficulty of each query or how much compute is needed to answer it correctly. To isolate and address this challenge, we adopt the standard Best-of-$N$ approach (Brown et al., 2024; Snell et al., 2024) for both compute counting (i.e., measuring the number of generations per query) and final output selection.

## 3 Methods

We present our approaches to solve the strategic test-time compute allocation problem introduced in Section 2.1. In Section 3.1, we first formulate test-time compute allocation as a bandit learning problem. We then introduce our algorithmic framework in Section 3.2, followed by extensions in Section 3.3 and theoretical analysis of compute efficiency in Section 3.4.

## 3.1 Test-time scaling as bandit learning

To address the challenge of strategic compute allocation under uncertainty, we introduce a novel bandit learning formulation tailored to LLM test-time compute objectives. Following the bandit terminology, we treat each query $x \in \mathcal{S}$ as an *action*, and interpret sampling action $x$ as allocating one unit of compute to query $x$ to obtain a randomly generated response $y$. After taking action $x$, the learner receives feedback from a reward oracle in the form of a score $r(x, y)$.

Our objective is to design an adaptive compute allocation algorithm that maximizes the fraction of queries that are correctly answered within a fixed compute budget $B$. Assuming availability of a sufficiently accurate reward oracle (e.g., ground truth labels), we approximate the correctness of a response using a user-specified threshold $\gamma \in [0, 1]$: a response $y$ to query $x$ is considered correct if $r(x, y) \ge \gamma$.[1] Formally, the algorithm adaptively distributes the total compute budget $B$ across all

---

[1] We assume access to a sufficiently accurate reward oracle in order to focus on the key challenge of adaptive compute allocation. This assumption is clearly satisfied in settings with ground truth labels, and is approximately satisfied by recently developed process reward models (Zhang et al., 2025a).

queries through an allocation $\{c(x_i)\}_{i=1}^n$, optimizing the following objective:

$$\max_{\{c(x_i)\}_{i=1}^n} \frac{1}{n} \sum_{i=1}^n \mathbb{I}\left(\max_{y \in g(x_i; c(x_i))} r(x_i, y) \geq \gamma\right),$$

where $g(x_i; c(x_i))$ denotes the set of $c(x_i)$ responses generated for query $x_i$.[2]

While our formulation is conceptually related to the bandit pure exploration problem (Bubeck et al., 2009; Jamieson & Nowak, 2014) and its thresholding bandit variants (Locatelli et al., 2016; Zhu et al., 2020), it fundamentally departs from the conventional objectives. Standard pure exploration settings aim to identify actions (queries) with high *expected* scores, which correspond—in our setting—to identifying a subset of easy queries that can be reliably answered by the LLM. In contrast, our objective aims at generating at least one high-quality (correct) response for each query, regardless of its expected score. To our knowledge, this not only introduces a novel bandit formulation but also opens the door to further exploration of bandit-based LLM test-time compute allocation.

## 3.2 OUR ALGORITHMIC FRAMEWORK

Based on the bandit formulation, we next present our algorithmic framework in Algorithm 1. Given a query set $\mathcal{S}$, Algorithm 1 initializes an *active set* $\mathcal{A} = \mathcal{S}$ that contains active queries that have not yet been confidently answered. For each query $x \in \mathcal{S}$, it maintains a response set $g(x)$, the best-scoring response $\check{y}(x)$ observed so far, and its corresponding reward score $\check{r}(x)$, as evaluated by the reward oracle $r$. Algorithm 1 proceeds in rounds, and operates based on two key components: an *exploration rule* and an *elimination rule*:

- **The exploration rule.** At each round, Algorithm 1 explores all queries in the active set, i.e., for each active query $x \in \mathcal{A}$, it generates $K$ new responses $\{y_i\}_{i=1}^K$ and updates the response set $g(x) \leftarrow g(x) \cup \{y_i\}_{i=1}^K$. We discuss extensions to this simple exploration rule in Section 3.3.

- **The elimination rule.** For each explored query $x$, let $y_{i^\star}$ denote the response that achieves the highest score among newly generated responses, i.e., $i^\star = \arg\max_{i \in [K]} r(x, y_i)$. If the reward $r(x, y_{i^\star})$ is greater than the previously observed best score $\check{r}(x)$, then Algorithm 1 (1) updates its maintained best-scoring response $\check{y}(x) = y_{i^\star}$ and the corresponding reward $\check{r}(x) = r(x, y_{i^\star})$; and (2) *eliminates* query $x$ from the active set $\mathcal{A}$ if the score $r(x, y_{i^\star})$ is also greater or equal to the elimination threshold $\gamma$.

Algorithm 1 terminates when the compute budget is exhausted (i.e., $B = 0$) or when all queries have been eliminated from the active set (i.e., $\mathcal{A} = \emptyset$). For each query $x \in \mathcal{S}$, Algorithm 1 outputs its maintained response set $g(x)$ for coverage evaluation, and its best-scoring response $\check{y}(x)$ for accuracy evaluation.

**Reward oracles.** Reward oracles have become a core component in test-time compute techniques, even for the vanilla uniform Best-of-$N$ algorithm (Brown et al., 2024; Snell et al., 2024). Common reward oracles include outcome reward models (ORMs, Cobbe et al. (2021)) and process reward models (PRMs, Uesato et al. (2022); Lightman et al. (2023); Zhang et al. (2025a)). For tasks with easy or automatic verification, such as math and code generation, ground truth (GT) labels can serve as an exact reward oracle. We emphasize that Algorithm 1 uses *the same number of reward oracle calls* as the uniform Best-of-$N$ algorithm, which relies on the reward oracle to select the final output.

**Hyperparameters.** Algorithm 1 takes two hyperparameters as input: the per-round per-query compute budget $K$ and a user-specified elimination threshold $\gamma$. The hyperparameter per-round per-query compute budget $K$ controls the granularity level of the budget allocation: a smaller value of $K$ leads to more fine-grained budget allocation with an increased number of allocation rounds. The elimination hyperparameter $\gamma$ decides when to eliminate a query from the active set $\mathcal{A}$. The value of $\gamma$ can be determined based on expert knowledge or based on cross-validation on a separate training set. These hyperparameters offer additional levels of flexibility for Algorithm 1. We conduct ablation studies of these hyperparameters in Section 4.4 and Appendix D.2.1.

---

[2]When the evaluation metric is Accuracy, one must further explicitly select and output the correct response.

---

**Algorithm 1** Strategic Test-Time Compute Allocation

---

**Input:** Query set $\mathcal{S}$, total compute budget $B$, reward oracle $r$, per-round per-query compute budget $K$, and elimination threshold $\gamma$.

1: For each query $x \in \mathcal{S}$, maintain a response set $g(x)$, the best-scoring response $\check{y}(x)$, and its associated reward $\check{r}(x)$.
2: Initialize the active set $\mathcal{A} \leftarrow \mathcal{S}$ to be the full query set.
3: **while** $B > 0$ and $|\mathcal{A}| > 0$ **do**
4:    **for** $x \in \mathcal{A}$ **do**
5:       Generate $K$ new responses $\{y_i\}_{i=1}^{K}$. Update $g(x) \leftarrow g(x) \cup \{y_i\}_{i=1}^{K}$ and $B \leftarrow B - K$. `// The exploration rule: allocating compute to all queries in the active set` $\mathcal{A}$`. We discuss extensions of the exploration rule in Section 3.3.`
6:       Get $i^\star \leftarrow \arg\max_{i \in [K]} r(x, y_i)$.
7:       **if** $r(x, y_{i^\star}) > \check{r}(x)$ **then**
8:          Update $\check{y}(x) \leftarrow y_{i^\star}$ and $\check{r}(x) \leftarrow r(x, y_{i^\star})$.
9:       **if** $r(x, y_{i^\star}) \geq \gamma$ **then**
10:      Update $\mathcal{A} \leftarrow \mathcal{A} \setminus \{x\}$.            `// The elimination rule.`
**Output:** For each $x \in \mathcal{S}$, output its response set $g(x)$ and the best-scoring response $\check{y}(x)$. `// Use` $g(x)$ `for coverage evaluation and` $\check{y}(x)$ `for accuracy evaluation.`

---

## 3.3 EXTENSIONS OF ALGORITHM 1

**Algorithm 1 with different aggregation strategies.** While our main discussion centers on Best-of-$N$, the proposed framework is flexible and can accommodate alternative aggregation strategies. Prior work (Wang et al., 2025a) has shown that Self-Consistency (SC) is often more effective for reasoning models—such as Qwen3-4B—due to their tendency to produce logically coherent outputs. To incorporate SC into Algorithm 1, we make two modifications: (1) the selection rule (line 8) now uses SC instead of a reward model (PRM), and (2) the elimination rule (line 10) is updated to eliminate a query once a certain proportion of its collected responses converge to the same answer (e.g., when over 50% agree). When using SC, the reliance on PRMs can be eliminated altogether. Experiments in Section 4.2 confirm that our algorithm remains effective when using SC as the aggregation rule.

**Algorithm 1 with different exploration rules.** While the base version of Algorithm 1 (ELIMINATION) explores all active queries uniformly at each round, our framework supports more targeted exploration strategies inspired by the pure exploration bandit literature. For example, Upper Confidence Bound (UCB) prioritizes queries with high empirical reward plus an uncertainty bonus (Kalyanakrishnan et al., 2012; Jamieson et al., 2014), while gap-based sampling (GAP) focuses on queries near the elimination threshold $\gamma$, allocating compute inversely proportional to the estimated reward gap (Locatelli et al., 2016). We also propose a novel entropy-based rule (ENTROPY) that selects queries with more diverse response patterns, as measured by empirical entropy, and encourages exploration of under-sampled queries. Experiments in Section 4.3 show that ENTROPY is particularly effective across extremely difficult query sets. We defer formulations of these strategies to Appendix B. For all the exploration rules, to prevent over-allocation of compute on difficult queries, we additionally introduce a per-query cap max_sample $\in \mathbb{N}$, which limits the number of generated responses for any individual query.

**Algorithm 1 with fine-grained token controls.** The default version of Algorithm 1 models compute cost as the number of response generations. However, it can be easily extended to track and control token-level usage. At each iteration, the algorithm can record token consumption and stop once the total token budget is reached. Alternatively, one can impose fine-grained token caps per generation. We evaluate this variant in Section 4.4 and find that Algorithm 1 continues to outperform baselines under the same token budget.

**Algorithm 1 with streaming queries.** In streaming settings, queries arrive sequentially, i.e., only the current query $x_t$ is accessible at round $t$. To adapt Algorithm 1 to this setting, we modify line 4 to focus solely on $x_t$ while keeping the rest of the framework unchanged. Also, to prevent over-allocation of compute on difficult queries, we additionally introduce a per-query cap max_sample $\in \mathbb{N}$, which limits the number of generated responses for any individual query. This constraint enforces a

local trade-off between exploration and exploitation and promotes balanced compute usage across the query stream. Ablation results in Section 4.4 show that this streaming variant remains competitive with our original method.

## 3.4 THEORETICAL INSIGHTS ON COMPUTE EFFICIENCY

A key strength of Algorithm 1 (and its variants in Section 3.3 with different exploration rules) lies in their ability to adapt compute based on estimated query difficulty: easier queries get fewer samples, while harder ones are given more when needed.

To better understand the advantage of Algorithm 1 over uniform compute allocation, we consider the following probabilistic model. For each query $x \in \mathcal{S}$, we model the correctness of the LLM's response in a single, independent generation as a Bernoulli random variable with parameter $\Delta_x \in (0, 1)$. That is, $X \sim \mathrm{Bernoulli}(\Delta_x)$, where $X = 1$ if the LLM answers the query correctly and $X = 0$ otherwise.

To ensure the reward oracle is compatible with this probability model and the specified threshold $\gamma$, we make the following assumption.

**Assumption 1.** *For any query $x \in \mathcal{S}$ and any randomly generated response $y$. $y$ correctly answers $x$ if and only if the reward oracle $r$ assigns a score $r(x, y) \geq \gamma$.*

Assumption 1 ensures that the elimination decision is aligned with the reward oracle and the threshold, allowing us to focus on the analysis of the adaptive design in Algorithm 1. This assumption is satisfied by the ground truth reward oracle, and holds approximately when the reward model is sufficiently accurate—a condition empirically supported by recent advances in high-quality process reward models (Zhang et al., 2025a).

Suppose $K = O(1)$, we derive the following quantitative comparison between Algorithm 1 and uniform compute allocation.

**Theorem 1.** *Assume Assumption 1, and fix any $\delta \in (0, 1)$. To output correct responses for all queries in $\mathcal{S}$ with probability at least $1 - \delta$, Algorithm 1 requires a total budget $B_{\mathsf{ours}} = \widetilde{\Theta}(\sum_{x \in \mathcal{S}} \frac{1}{\Delta_x})$. This matches the information-theoretic lower bound up to logarithmic factors. In contrast, a uniform allocation strategy requires budget $B_{\mathsf{unif}} = \widetilde{\Omega}(\max_{x \in \mathcal{S}} \frac{|\mathcal{S}|}{\Delta_x})$ to achieve the same guarantee.*

Theorem 1 highlights the efficiency advantage of Algorithm 1 over uniform allocation. In particular, Algorithm 1 requires a total budget of order $\widetilde{\Theta}(\sum_{x \in \mathcal{S}} \frac{1}{\Delta_x})$, whereas uniform allocation requires $\widetilde{\Omega}(\max_{x \in \mathcal{S}} \frac{|\mathcal{S}|}{\Delta_x})$. Thus, by adapting to query difficulty, Algorithm 1 can significantly reduce total compute. For example, if $|\mathcal{S}| = n$ and $\Delta_{x_i} = i/n$, then $B_{\mathsf{ours}} = \widetilde{O}(n)$ while $B_{\mathsf{unif}} = \widetilde{\Omega}(n^2)$, so uniform allocation can be a factor of $n$ less efficient.

One may further consider the classical two-stage explore-then-commit (ETC) strategy, which allocates a fixed exploration budget to each query before committing additional compute based on the observed outcomes (Lattimore & Szepesvári, 2020). Despite its partial adaptivity, ETC remains fundamentally limited because its first stage performs uniform allocation. We next show that this structural constraint already leads to strictly larger budget complexity in heterogeneous regimes.

**Proposition 1.** *Assume Assumption 1, and fix any $\delta \in (0, 1)$. Consider any two-stage explore-then-commit algorithm that allocates a fixed exploration budget $m$ to every query in the first stage. To output correct responses for all queries in $\mathcal{S}$ with probability at least $1 - \delta$, any such algorithm must incur total compute $B_{\mathsf{ETC}} = \widetilde{\Omega}(\sum_{x \in \mathcal{S}} \max(m, \frac{1}{\Delta_x}))$.*

In particular, when $m$ is nontrivial and many queries are easy (i.e., when $\frac{1}{\Delta_x}$ is small), ETC incurs an unavoidable uniform-exploration overhead of $\Theta(m)$ per easy query, whereas Algorithm 1 only spends $\widetilde{O}(1/\Delta_x)$. This establishes a strict separation between ETC and our fully adaptive allocation. Consistent with this theory, our empirical results in Section 4 also show that the ETC baseline (denoted as TWO STAGE) is substantially outperformed by our algorithm.

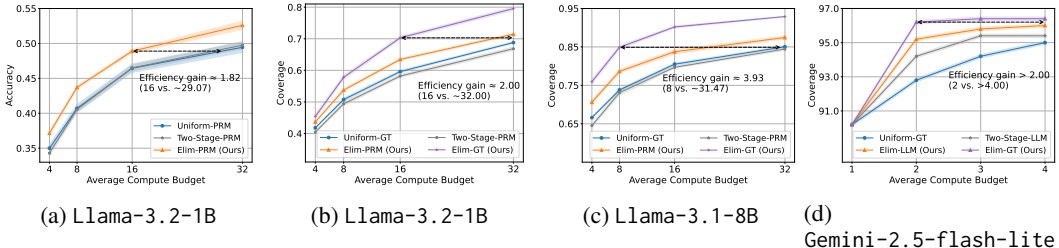

(a) `Llama-3.2-1B`  (b) `Llama-3.2-1B`  (c) `Llama-3.1-8B`  (d) `Gemini-2.5-flash-lite`

Figure 2: Accuracy and coverage comparisons on MATH-500 across models of different sizes. Accuracy results of `Llama-3.1-8B-Instruct` and `Gemini-2.5-flash-lite` are presented in Fig. 1.

## 4 EXPERIMENTS

We describe experimental setups in Section 4.1, present main results in Section 4.2, offer further analysis in Section 4.3, and report ablations in Section 4.4. Additional experimental details and results are deferred to Appendix D.

### 4.1 EXPERIMENTAL SETUP

**Datasets.** We examine the performance of our algorithms on standard math and code benchmarks: MATH-500 and AIME25 (Lightman et al., 2023; Hendrycks et al., 2021; AIME, 2025) and Live-CodeBench (Jain et al., 2024). MATH-500 contains 500 math questions, AIME25 contains 30 difficult math questions, and the LiveCodeBench contains 479 code execution questions that were collected from 5/1/2023 to 12/1/2023. From MATH-500, we further construct one challenging subset: MATH-500-Hard-16, which contain questions that cannot be correctly answered after allocating 16 units of compute. Intuitively, this subset consists of the most difficult queries in the MATH-500 dataset.

**Baselines.** We compare our algorithms with the uniform Best-of-$N$ baseline (Brown et al., 2024), referred to as UNIFORM, and a two-stage baseline, referred to as TWO STAGE (Damani et al., 2024; Wang et al., 2025b). The TWO STAGE baseline first uniformly allocates compute to estimate problem difficulty (stage 1) and then allocates the remaining compute proportionally (stage 2); in other words, it operates in an explore-then-commit style. For the TWO STAGE algorithm, we vary the stage 1 compute ratio from $\{25\%, 50\%, 75\%\}$ and report the best results. We report the performance of our Algorithm 1 (ELIMINATION) and its variants introduced in Section 3.3.

**Models and metrics.** We conduct experiments with commonly used LLMs of various sizes, including `Llama-3.2-1B-Instruct` and `Llama-3.1-8B-Instruct` (Grattafiori et al., 2024), as well as more recently developed reasoning models `DeepSeek-R1-Distill-Llama-8B` (DeepSeek-AI et al., 2025), `Qwen3-4B` (Yang et al., 2025), and `Gemini-2.5-Flash-Lite` (Comanici et al., 2025). For MATH-500, we use different reward oracles depending on the base model family: for Llama models, we use the PRM `Qwen2.5-Math-PRM-7B` (Zhang et al., 2025a); for `Gemini-2.5-Flash-Lite`, we use an LLM-as-a-judge oracle based on `Gemini-3.0-Flash`. In addition, we also report results under the ground-truth (GT) oracle when available.[3] For AIME25, we use the Self-Consistency (SC) variants of baselines and our methods, as recent work shows that SC is more effective for reasoning models (Wang et al., 2025a). For LiveCodeBench, since correctness can be deterministically verified by code execution, we use the GT reward oracle and report only the coverage metric (equivalent to accuracy). We conduct experiments under average compute budgets of $\{4, 8, 16, 32\}$ and report results averaged over 4 random runs, with shaded regions in plots representing $\pm 0.5$ standard deviations.

### 4.2 MAIN RESULTS

**MATH-500 results.** Fig. 2 presents experimental results on the MATH-500 dataset across two LLMs and two evaluation metrics (the accuracy result of `Llama-3.1-8B-Instruct` on MATH-500 is pre-

---

[3]When using the GT reward oracle, accuracy and coverage are equivalent. In this case, we only report coverage.

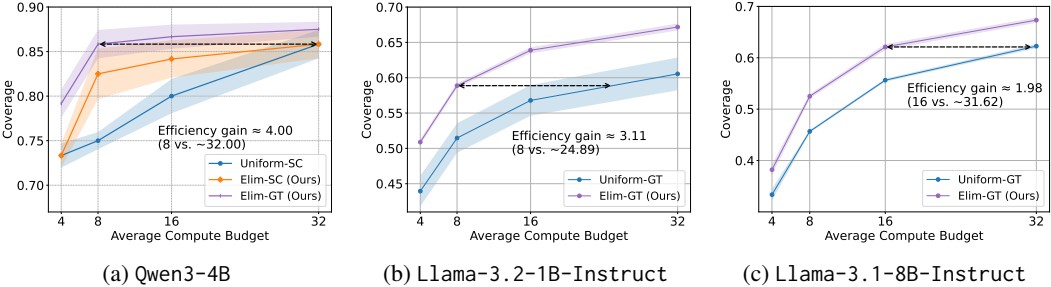

Figure 3: Results on AIME25 (*left*) and LiveCodeBench (*middle and right*). Accuracy result of `Qwen3-4B` on AIME25 and coverage result of `DeepSeek-R1-Distill-Llama-8B` on LiveCodeBench are presented in Fig. 1.

sented in the first plot of Fig. 1). Across all configurations, all variants of our Algorithm 1 consistently outperform both baselines. Under the accuracy metric, when the average compute budget is 16, our method achieves a 2.50% absolute improvement (7.37% relative) on `Llama-3.2-1B-Instruct`; this corresponds to a $1.82\times$ efficiency gain as shown on plot (a): UNIFORM takes $1.82\times$ compute to achieve the same performance. For `Llama-3.1-8B-Instruct`, we observe a 1.40% absolute improvement (4.11% relative), with a $2\times$ efficiency gain (Fig. 1, first plot). For the coverage metric, when the average compute budget is 16, our method yields a 10.70% absolute improvement (17.95% relative) on `Llama-3.2-1B-Instruct`, resulting in a $2\times$ efficiency gain (Fig. 2, plot (b)).[4] When the average budget is 8, we observe an 11.10% absolute gain (15.04% relative) on `Llama-3.1-8B-Instruct`, yielding a $3.93\times$ efficiency gain (Fig. 2, plot (c)).

We further evaluate our method on the frontier model `Gemini-2.5-Flash-Lite` with reasoning enabled. Given its strong base performance, we use smaller average compute budgets of $\{1, 2, 3, 4\}$. Because existing PRMs are not reliable at this performance level, we instead use `Gemini-3.0-Flash` as an LLM-as-a-judge oracle for `Gemini-2.5-Flash-Lite`. Under the accuracy metric, our method improves performance by 3.4% absolute (3.66% relative), corresponding to a $> 2.00\times$ efficiency gain (Fig. 1, second plot). Under the coverage metric, we obtain a 2.4% absolute improvement (2.64% relative), again exceeding a $2\times$ efficiency gain (Fig. 2, plot (d)).

**AIME25 results.** Fig. 3 reports results on the AIME25 dataset with `Qwen3-4B` (accuracy result of `Qwen3-4B` on AIME25 is presented at the third plot of Fig. 1). Following Yang et al. (2025), we enable the reasoning ability of `Qwen3-4B` and set the max token length to 38,912. Under the accuracy metric (Fig. 1, third plot), we are able to gain an absolute of 3.33% performance gain (4.60% relative) when the average compute budget is 8 on `Qwen3-4B`, yielding a $4.00\times$ efficiency gain. For the coverage metric (Fig. 3, left plot), when the average compute budget is 16, we observe a 10.82% absolute gain (14.44% relative) on `Qwen3-4B`, yielding a $4.00\times$ efficiency gain.

**LiveCodeBench results.** Fig. 3 (middle and right) presents results on the LiveCodeBench with `Llama` models of different sizes, and the fourth plot of Fig. 1 presents results of `DeepSeek-R1-Distill-Llama-8B`. As described in Section 4.1, we use the GT reward oracle and report coverage, which is equivalent to accuracy in this setting. We report results for the ELIMINATION variant only, as UCB and GAP behave identically to ELIMINATION under the GT oracle. Across all compute budgets, our method consistently outperforms uniform allocation. With an average compute budget of 16, `Llama-3.2-1B-Instruct` achieves a 6.47% absolute improvement (11.63% relative), corresponding to a $1.98\times$ efficiency gain (Fig. 3, middle plot). With an average compute budget of 8, `Llama-3.1-8B-Instruct` achieves a 7.41% absolute improvement (14.40% relative), corresponding to a $3.11\times$ efficiency gain (Fig. 3, right plot). With an average compute budget of 8, `DeepSeek-R1-Distill-Llama-8B` achieves a 9.97% absolute improvement (12.30% relative), corresponding to a $4.00\times$ efficiency gain (Fig. 1, fourth plot).

**MATH-500-Hard result.** Fig. 4 (left) presents experimental results on the MATH-500-Hard-16 dataset, which was constructed to include the most challenging questions in the MATH-500

---

[4]Under the GT reward oracle, we report only the performance of ELIMINATION, as other variants yield similar results. See Appendix D.2.1 for full comparisons.

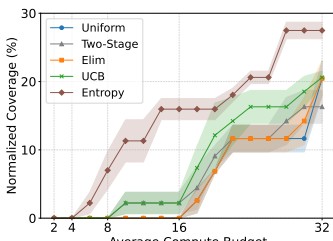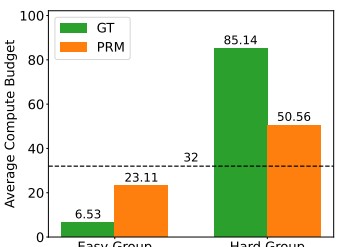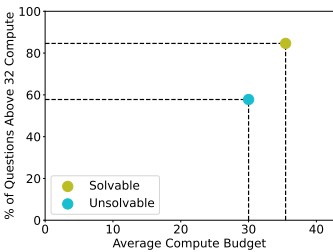

Figure 4: Results on MATH-500-Hard with `Llama-3.2-1B-Instruct` and an average compute budget of 32. *Left:* Coverage result on MATH-500-HARD-16. *Center:* Allocation behavior of Algorithm 1 for easy vs. hard groups. *Right:* Allocation behavior of ENTROPY for solvable vs. unsolvable groups.

benchmark. We evaluate performance using the GT reward oracle, as PRM-based scores are less reliable on these difficult questions. On this dataset, baseline methods and the vanilla version of our Algorithm 1 (ELIMINATION) doesn't performance well, as many questions can not be solved by the base LLM. However, we find alternative exploration rules introduced in Section 3.3—particularly ENTROPY—achieve significantly better results. These findings highlight the benefits of incorporating more nuanced exploration strategies, such as those developed in Section 3.3, for effective compute allocation on challenging benchmarks. We defer more experiment results to Appendix D.2.2.

### 4.3 ANALYSIS ON THE ADVANTAGES OF STRATEGIC COMPUTE ALLOCATION

We conduct further empirical analyses to illustrate the benefits of strategic compute allocation in two settings: (1) on standard datasets containing both easy and hard queries, and (2) on challenging datasets containing both solvable and unsolvable queries. All experiments are conducted using `Llama-3.2-1B-Instruct` with an average compute budget of 32.

**Strategic allocation on standard datasets.** In the first analysis, we partition the MATH-500 dataset into two subsets: queries that can be correctly answered with at most 32 units of compute (easy group), and those that cannot (hard group). Intuitively, the easy group consists of questions that require less than 32 units of compute to solve, while the hard group includes questions that would benefit from additional compute. In the middle plot of Fig. 4, we visualize the compute allocation of Algorithm 1 under both PRM and GT reward oracles. Compared to uniform allocation, our algorithm allocates fewer resources to easy queries and more to hard ones. This demonstrates the ability of Algorithm 1 to strategically allocate compute—reserving effort for harder queries that need it most.

**Strategic allocation on challenging datasets.** In the second analysis, we consider the MATH-500-Hard-16 dataset and divide it into solvable queries and unsolvable ones, where the latter cannot be correctly answered even after allocating 500 units of compute. In such settings, effective allocation should prioritize the solvable subset, as investing in unsolvable queries leads to wasted compute. The right plot of Fig. 4 shows that under a 32-unit compute budget, ENTROPY allocates more compute on average to solvable queries, and a larger fraction of them receive more than 32 samples. This demonstrates that our method learns to concentrate compute on tractable instances, avoiding waste on queries unlikely to be resolved.

To understand why ENTROPY behaves this way, we inspect model outputs on these challenging questions in detail. We observe that unsolvable queries often yield invalid responses (e.g., incomplete or poorly formatted), leading to lower entropy across generations. In contrast, solvable queries tend to produce more diverse and well-formed outputs, resulting in higher entropy. We defer detailed experiments and analysis of ENTROPY extension to Appendix D.2.2.

### 4.4 ABLATION STUDIES

**Algorithm 1 with different $K$.** All main experiments in Section 4.2 use the default setting $K = 1$. Smaller values of $K$ enable finer-grained adaptive allocation and are generally preferred for maximizing performance. In Fig. 5 (left), we conduct an ablation study with $K \in \{1, 2, 4, 8\}$ on the

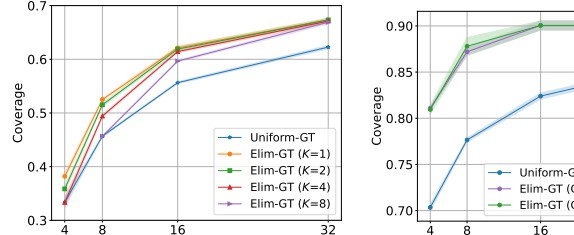 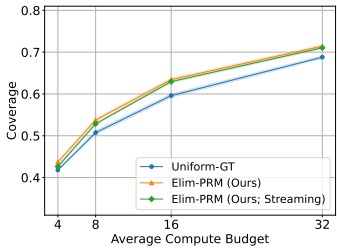

Figure 5: Ablation studies for Algorithm 1. *Left:* Effect of $K$ on LiveCodeBench with `Llama-3.2-1B-Instruct`. *Center:* Performance of the token-controlled variant on LiveCodeBench with `DeepSeek-R1-Distill-Llama-8B`. *Right:* Performance of the streaming variant on MATH-500 with `Llama-3.2-1B-Instruct`.

LiveCodeBench dataset using `Llama-3.2-1B-Instruct`. Across all values of $K$, our method consistently outperforms the uniform baseline. While larger $K$ reduces allocation granularity—making performance closer to uniform allocation under tight budgets—the gap narrows as the average compute budget increases. These results show that Algorithm 1 is robust to the choice of $K$.

**Algorithm 1 with token controls.** We evaluate Algorithm 1 in a token-controlled setting, where compute is measured by total token usage rather than the number of generations. To ensure comparability, we match the average token budget used by uniform allocation and discard excess samples when needed. As shown in Fig. 5 (middle), on the LiveCodeBench dataset with `DeepSeek-R1-Distill-Llama-8B`, Algorithm 1 still outperforms the uniform baseline even under equivalent token budgets.

**Algorithm 1 with streaming queries.** We also test Algorithm 1 in a streaming setting, where queries arrive sequentially and the full query pool is not available in advance. Using the variant described in Section 3.3, we evaluate performance in this setting and report results in Fig. 5 (right). Our method performs comparably to its pool-based counterpart and still outperforms UNIFORM with pool access, demonstrating its effectiveness under streaming constraints.

## 5 CONCLUSION

We introduce a new perspective on LLM test-time scaling by formulating strategic compute allocation as a bandit learning problem. We develop adaptive algorithms that estimate query difficulty on the fly and allocate compute to maximize the fraction of correctly answered queries under a fixed compute budget. Our framework is flexible and extends naturally to incorporate alternative aggregation and exploration strategies, as well as to support both streaming and token-constrained settings. We provide theoretical guarantees that strategic compute allocation improves compute efficiency over uniform allocation, and we empirically demonstrate substantial performance improvements—up to 11.10% on MATH-500, 10.82% on AIME25, and 11.23% on LiveCodeBench. These findings underscore the potential of bandit-based compute allocation for more effective test-time scaling.

ACKNOWLEDGMENTS

YZ acknowledges support from NSF IIS 2425006.

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

# A    RELATED WORK

**Test-time compute techniques.**    Scaling test-time compute (TTC) has emerged as a powerful class of methods for improving the performance of large language models, typically without requiring additional parameter updates. In-context learning (Brown et al., 2020), including its scaling to many-shot regimes (Agarwal et al., 2024; Bertsch et al., 2024), as well as prompting- or search-based methods such as Chain-of-Thought (Wei et al., 2023) and Tree-of-Thought (Yao et al., 2023; Feng et al., 2023), have demonstrated that carefully designed test-time techniques can match or even surpass finetuned models (Mosbach et al., 2023). Self-reflection (Madaan et al., 2023) is another popular technique for leveraging TTC to improve performance: by prompting the LLM to iteratively refine its own generations, the model can produce higher-quality responses across a range of tasks (Chen et al., 2023; Gou et al., 2023). Muennighoff et al. (2025) further demonstrates that simply increasing the number of generated "thinking" tokens leads to substantial performance gains.

Repeated sampling methods—most notably Best-of-$N$ (Brown et al., 2024; Snell et al., 2024; Wang et al., 2022)—have become popular for scaling test-time compute, especially when combined with high-quality reward models (Cobbe et al., 2021; Uesato et al., 2022; Lightman et al., 2023; Zhang et al., 2025a). Building on this line of work, recent—and in some cases concurrent—efforts have proposed adaptive variants of Best-of-$N$ that dynamically allocate compute for a given query (Sun et al., 2024; Manvi et al., 2024; Tan et al., 2025). However, these methods focus on adaptive allocation *within an individual query*, without considering opportunities to redistribute compute across a set of queries. In contrast, we study *strategic compute allocation across multiple queries*, introducing an additional layer of optimization—for example, deciding when to transfer unused budget from easier queries to harder ones. The problem setting in Damani et al. (2024) is closely related, as they also consider multi-query compute allocation. However, their approach relies on a two-stage schedule that requires training an additional model in the first stage to guide compute distribution, incurring extra compute cost. In contrast, we formulate the problem as a novel bandit learning task and develop *fully adaptive* algorithms that learn to allocate compute *on the fly*, without any additional training overhead. Furthermore, we provide the first theoretical result that provably demonstrates the advantage of strategic test-time compute allocation over uniform allocation.

**Bandit learning and pure exploration.**    Bandit learning is a fundamental framework for sequential decision making under uncertainty, where an agent must choose among a set of actions (or arms) to optimize a long-term objective with limited feedback (Bubeck & Cesa-Bianchi, 2012; Lattimore & Szepesvári, 2020). Popular algorithms include Upper Confidence Bound (UCB, Auer et al. (2002); Audibert & Bubeck (2009); Chu et al. (2011); Zhu & Nowak (2020, 2022); Garivier et al. (2022)), which selects the action with the highest upper confidence bound; Thompson Sampling (Thompson, 1933; Chapelle & Li, 2011; Agrawal & Goyal, 2012; Russo et al., 2018), which selects the action with the highest sampled reward from the posterior; and inverse gap weighting strategies (Foster & Rakhlin, 2020; Foster et al., 2021; Zhu et al., 2022a; Zhu & Mineiro, 2022; Rucker et al., 2023), which sample actions with probabilities inversely proportional to their estimated reward gaps. Bandit algorithms have been widely applied in domains such as online recommendation systems (Li et al., 2010), clinical trials (Villar et al., 2015), hyperparameter tuning (Li et al., 2018), and more recently applications with LLMs (Shi et al., 2024; Chen et al., 2024).

Pure exploration (Bubeck et al., 2009; Jamieson & Nowak, 2014), also known as the best arm identification (BAI) problem, is a key subfield of bandit learning that aims to identify high-performing arms using as few samples as possible. Core algorithms include successive elimination (Even-Dar et al., 2002, 2006; Karnin et al., 2013), UCB-based strategies (Kalyanakrishnan et al., 2012; Kaufmann & Kalyanakrishnan, 2013; Jamieson et al., 2014), and gap-based sampling methods (Locatelli et al., 2016). Recent extensions generalize these techniques to more expressive function classes, including linear models (Fiez et al., 2019; Katz-Samuels et al., 2020; Zhu et al., 2022b), kernel functions (Du et al., 2021), and neural networks (Zhu et al., 2021). Our work introduces a novel pure-exploration-style bandit formulation, tailored to LLM test-time compute allocation—a setting not previously explored in this context. We treat each query as a bandit action and adaptively allocate compute to maximize the fraction of queries correctly answered under a fixed compute budget. This formulation enables the use of classical bandit techniques such as elimination rules, confidence bounds, and gap-based sampling. In addition, we propose a new entropy-based sampling strategy (Section 3.3) that prioritizes queries with diverse response patterns. While our formulation is

conceptually related to the thresholding bandit problem and its variants (Locatelli et al., 2016; Zhu & Nowak, 2020), it departs fundamentally in its objective. Thresholding bandits aim to identify actions (queries) whose *expected reward* exceeding a given threshold. In contrast, our goal is to generate at least one high-quality response for each query, regardless of its expected score.

## B   ALGORITHM 1 WITH DIFFERENT EXPLORATION RULES

Our main algorithmic framework (Algorithm 1) is presented with a simple exploration rule that explores all queries within the active set (lines 4-5). In practice, this rule can be flexibly extended to incorporate diverse exploration objectives. Motivated by developments in the bandit pure exploration literature, we introduce several alternative exploration rules in the following. We use $g(x)$ to denote the response set to query $x$, and $N(x) := |g(x)|$ to denote the number of generations so far.

- **Upper confidence bound (UCB).** For any active query $x \in \mathcal{A}$, let $\widehat{r}(x) := \sum_{y_i \in g(x)} r(x, y_i)/N(x)$ denote the empirical average reward based on previously collected responses. Let $\lambda > 0$ be a hyperparameter. At each round, the UCB exploration rule selects the query based on the following criteria:

$$\arg\max_{x \in \mathcal{A}} \ \widehat{r}(x) + \lambda N(x)^{-1/2}.$$

  This exploration rule follows the principle of optimism in the face of uncertainty (Kalyanakrishnan et al., 2012; Jamieson et al., 2014), and prioritizes on selecting queries that are more likely to be solved (i.e., those with higher average rewards). The term $\lambda N(x)^{-1/2}$ is used to construct the upper confidence bound of the reward.

- **GAP.** For any active query $x \in \mathcal{A}$, let $\widehat{r}(x) := \sum_{y_i \in g(x)} r(x, y_i)/N(x)$ denote the empirical average reward based on previously collected responses. At each round, the GAP exploration rule selects the query based on the following criterion:

$$\arg\min_{x \in \mathcal{A}} \ (\gamma - \widehat{r}(x)) \cdot N(x)^{-1/2}.$$

  This exploration rule prioritizes queries whose estimated reward is close to the elimination threshold $\gamma$, with a preference toward less-explored queries. The weighting term $N(x)^{-1/2}$ ensures that compute is allocated inversely proportional to the reward gap from the elimination threshold (Locatelli et al., 2016).

- **ENTROPY.** For any active query $x \in \mathcal{A}$, let $\{v_k\}$ be the set of distinct responses in $g(x)$, and define the empirical probability of observing response $v_k$ as $p_k(x) := |\{i : y_i = v_k, y_i \in g(x)\}|/N(x)$. Let $H(x) = -\sum_k p_k(x) \log p_k(x)$ denote the entropy of the empirical response distribution $p(x)$. Let $\lambda > 0$ be a hyperparameter. At each round, the ENTROPY exploration rule selects the query based on the following criterion:

$$\arg\max_{x \in \mathcal{A}} \ H(x) + \lambda N(x)^{-1/2}.$$

  This exploration rule, proposed in our work, prioritizes queries that elicit a more diverse set of responses, as indicated by higher entropy. The term $\lambda N(x)^{-1/2}$ encourages exploration of under-explored queries by balancing the trade-off between response diversity and sample count.

## C   SUPPORTING RESULTS FROM SECTION 3.4

### C.1   PROOF OF THEOREM 1

**Theorem 1.** *Assume Assumption 1, and fix any $\delta \in (0, 1)$. To output correct responses for all queries in $\mathcal{S}$ with probability at least $1 - \delta$, Algorithm 1 requires a total budget $B_{\mathsf{ours}} = \widetilde{\Theta}(\sum_{x \in \mathcal{S}} \frac{1}{\Delta_x})$. This matches the information-theoretic lower bound up to logarithmic factors. In contrast, a uniform allocation strategy requires budget $B_{\mathsf{unif}} = \widetilde{\Omega}(\max_{x \in \mathcal{S}} \frac{|\mathcal{S}|}{\Delta_x})$ to achieve the same guarantee.*

*Proof.* We first prove that $B_{\text{ours}} = \widetilde{\Theta}(\sum_{x \in \mathcal{S}} \frac{1}{\Delta_x})$ under the elimination rule of Algorithm 1. Note that under Assumption 1, a query is eliminated if and only if it is correctly answered.[5]

For the upper bound, we denote $\bar{\delta} := \delta/|\mathcal{S}|$, $n_x := \frac{1}{\Delta_x} \log \frac{1}{\bar{\delta}}$ and consider the following event:

$$E_x := \{\text{query } x \text{ will be correctly answered within } n_x \text{ random generations}\}.$$

We know that $E_x$ happens with probability at least $1 - \bar{\delta}$ as the probability of $\overline{E}_x$ is upper bounded by $\bar{\delta}$:

$$(1 - \Delta_x)^{n_x} \le e^{-\Delta_x \cdot n_x} = e^{-\Delta_x \cdot \frac{1}{\Delta_x} \log \frac{1}{\bar{\delta}}} = \bar{\delta}.$$

A union bound over $x \in \mathcal{S}$ leads to $\mathbb{P}(\cup_{x \in \mathcal{S}} E_x) \ge 1 - \sum_{x \in \mathcal{S}} \mathbb{P}(\overline{E}_x) \ge 1 - \delta$. As a result, the with probability of at least $1 - \delta$, Algorithm 1 and its variants in Appendix B output correct responses for all queries with compute budget $\sum_{x \in \mathcal{S}} n_x = \sum_{x \in \mathcal{S}} \frac{1}{\Delta_x} \log \frac{|\mathcal{S}|}{\delta} = \widetilde{O}(\sum_{x \in \mathcal{S}} \frac{1}{\Delta_x})$.

For the lower bound, we quantify the amount of compute $n_x$ needed to correctly answer any fixed query $x \in \mathcal{S}$ with probability at least $1 - \delta$. Recall that each independent generation answers $x$ correctly with probability $\Delta_x \in (0, 1)$. Therefore, after $n_x$ independent generations, the probability that *none* of them is correct equals

$$\mathbb{P}(\text{query } x \text{ is not correctly answered within } n_x \text{ generations}) = (1 - \Delta_x)^{n_x}.$$

If an algorithm outputs a correct response for *all* queries in $\mathcal{S}$ with probability at least $1 - \delta$, then in particular it must output a correct response for this fixed query $x$ with probability at least $1 - \delta$, which requires $(1 - \Delta_x)^{n_x} \le \delta$. Taking logarithms gives $n_x \ge \frac{\log(1/\delta)}{-\log(1-\Delta_x)}$. Using the inequality $-\log(1-u) \le \frac{u}{1-u}$ for all $u \in (0, 1)$, we further obtain

$$n_x \ge \frac{1 - \Delta_x}{\Delta_x} \log \frac{1}{\delta}.$$

Summing the above inequality over all $x \in \mathcal{S}$ yields the compute lower bound $\Omega(\sum_{x \in \mathcal{S}} \frac{1-\Delta_x}{\Delta_x} \log \frac{1}{\delta})$. In particular, if $\Delta_x \le 1 - c$ for all $x \in \mathcal{S}$ and some fixed constant $c \in (0, 1)$, then the lower bound simplifies to $\widetilde{\Omega}(\sum_{x \in \mathcal{S}} \frac{1}{\Delta_x})$, which matches our upper bound up to logarithmic factors.

As for uniform compute allocation, it allocates the same amount of compute for each query $x \in \mathcal{S}$. We thus only need to quantify the amount of compute $\bar{n}$ needed to correctly answer $\bar{x} := \arg\min_{x \in \mathcal{S}} \Delta_x$, the hardest query within $\mathcal{S}$, with high probability. The above lower bound analysis shows that one needs $\Omega(\frac{1}{\Delta_{\bar{x}}} \log \frac{1}{\delta})$ compute to correctly answer the $\bar{x}$ with probability at least $1 - \delta$. Since uniform allocation assigns the same budget to every query, the total compute must satisfy $B_{\text{unif}} = \Omega(\frac{|\mathcal{S}|}{\Delta_{\bar{x}}} \log \frac{1}{\delta}) = \widetilde{\Omega}(\max_{x \in \mathcal{S}} \frac{|\mathcal{S}|}{\Delta_x})$. $\qquad\square$

### C.2 PROOF OF PROPOSITION 1

**Proposition 1.** *Assume Assumption 1, and fix any $\delta \in (0, 1)$. Consider any two-stage explore-then-commit algorithm that allocates a fixed exploration budget $m$ to every query in the first stage. To output correct responses for all queries in $\mathcal{S}$ with probability at least $1 - \delta$, any such algorithm must incur total compute $B_{\text{ETC}} = \widetilde{\Omega}(\sum_{x \in \mathcal{S}} \max(m, \frac{1}{\Delta_x}))$.*

*Proof.* Suppose the ETC algorithm allocates $m$ generations to every query $x \in \mathcal{S}$ in the exploration stage. Let $n_x$ denote the *total* number of generations allocated to query $x$ across both stages. Then we have $n_x \ge m$ for all $x \in \mathcal{S}$. On the other hand, by the same per-query lower bound argument used in the proof of Theorem 1, to correctly answer query $x$ with probability at least $1 - \delta$ it is necessary that $n_x = \Omega(\frac{1}{\Delta_x} \log \frac{1}{\delta})$. Combining these two necessary conditions yields $n_x = \Omega(\max(m, \frac{1}{\Delta_x} \log \frac{1}{\delta}))$. Summing over all $x \in \mathcal{S}$ gives $B_{\text{ETC}} = \Omega(\sum_{x \in \mathcal{S}} \max(m, \frac{1}{\Delta_x} \log \frac{1}{\delta})) = \widetilde{\Omega}(\sum_{x \in \mathcal{S}} \max(m, \frac{1}{\Delta_x}))$. $\qquad\square$

---

[5]Since the elimination rule works for all variants of Algorithm 1 introduced in Appendix B, the guarantee also holds for these variants.

# D    OTHER DETAILS AND RESULTS FOR EXPERIMENTS

## D.1    ADDITIONAL DETAILS ON EXPERIMENTAL SETUPS

### D.1.1    ADDITIONAL HYPERPARAMETERS

We conduct all experiments on two NVIDIA RTX 6000 Ada GPUs. We use vLLM (Kwon et al., 2023) for LLM response generation, with a temperature of $0.6$.

The extended exploration rules introduced in Appendix B require additional hyperparameters. For UCB, we set $\lambda = 1$; for ENTROPY, we set $\lambda = 3$. For UCB, GAP, and ENTROPY, we additionally set a `max_samples` hyperparameter to prevent over-allocation of compute on difficult queries. The values of the `max_samples` hyperparameter will be introduced along with the experiment results.

### D.1.2    ADDITIONAL DETAILS ON MATH-500-HARD DATASETS

As discussed in Section 4.1, we construct the MATH-500-Hard dataset by removing queries from MATH-500 that can be solved with 16 units of compute. On top of this, we construct another dataset by removing queries that can be solved with 8 units of compute, and we call this dataset MATH-500-Hard-8. After removing these relatively easy queries, MATH-500-Hard-8 contains 71 challenging queries, and MATH-500-Hard-16 contains 56 challenging queries. We further divide MATH-500-Hard queries into two subsets: the subset that cannot be solved after allocating $M$ compute units (Unsolvable) and the rest (Solvable). We set $M = 500$ for `Llama-3.2-1B-Instruct`, and $M = 350$ for `Llama-3.1-8B-Instruct`. For normalized coverage, we calculate it based on the percentage of solved questions over all solvable questions. On these MATH-500-Hard datasets, we select the best from `max_samples` $\in \{36, 48, 64\}$ for UCB, GAP, and ENTROPY. The setting of `max_samples` is discussed in the same way as Section 3.3.

### D.1.3    PROMPTS SELECTION

**Prompts for MATH-500.**    We include four in-context examples in the prompt for MATH-500 for our Llama models. An illustrative example used in our experiments is shown below.

> **MATH prompts**
>
> **Problem:** If $\det \mathbf{A} = 2$ and $\det \mathbf{B} = 12$, then find $\det(\mathbf{A}\,\mathbf{B})$.
>
> **Solution:** We have that
> $$\det(\mathbf{A}\,\mathbf{B}) = \det(\mathbf{A})\,\det(\mathbf{B}) = (2)\,(12) = \boxed{24}.$$
> Final Answer: The final answer is $24$. I hope it is correct.
>
> $$\vdots$$
>
> **Problem: {actual test question}**
> **Solution:**

**Prompts for LiveCodeBench.**    We use the prompt provided on the official GitHub of Live-CodeBench (Jain et al., 2024). The prompt used in our experiments is shown below.

---

**LiveCodeBench CoT prompt**

You are given a Python function and an assertion containing an input to the function. Complete the assertion with a literal (no unsimplified expressions, no function calls) containing the output when executing the provided code on the given input, even if the function is incorrect or incomplete. Do NOT output any extra information. Execute the program step by step before arriving at an answer, and provide the full assertion with the correct output in [ANSWER] and [/ANSWER] tags, following the examples.
[PYTHON] def performOperation(s): s = s + s return "b" + s + "a" assert performOperation(s = "hi") == ?? [/PYTHON]
[THOUGHT] Let's execute the code step by step: 1. The function `performOperation` is defined, which takes a single argument `s`. 2. The function is called with the argument `"hi"`, so within the function, `s` is initially `"hi"`. 3. Inside the function, `s` is concatenated with itself, so `s` becomes `"hihi"`. 4. The function then returns a new string that starts with `"b"`, followed by `s` (now `"hihi"`), and ends with `"a"`. 5. The return value is therefore `"bhihia"`. [/THOUGHT]
[ANSWER] assert performOperation(s = "hi") == "bhihia" [/ANSWER]

---

**AIME prompt.** The same format as MATH-500 prompts, but without any in-context examples.

---

**AIME prompt example**

**Problem: {actual test question}**

**Solution:**

---

**Output segmentation for PRMs.** The output from our models follow a structured format shown below. We segment each solution into individual reasoning steps using the marker `Step ##`, and feed the resulting step-level segments to the PRM `Qwen2.5-Math-PRM-7B` for evaluation. Prior work commonly uses `\n\n` as a step delimiter and finds it effective (Zhang et al., 2025b). In our setting, however, we find that `Step ##` better matches the formatting of our outputs and yields higher overall performance; we therefore adopt `Step ##` for segmentation throughout (see Table 1).

---

**Expected generation from both Llama models about MATH questions**

Step 1: [Description of first step]
Step 2: [Description of second step]
Step 3: [Description of third step]
Step...
The final answer is: $\boxed{}$

---

Table 1: Effects of segmentation keywords on MATH-500 with `Llama-3.1-8B-Instruct`

| Method | 4 | 8 | 16 | 32 |
|---|---|---|---|---|
| Uniform (delimiter: ##) | 60.26% | 63.30% | 66.56% | 67.80% |
| Uniform (delimiter: \n\n) | 59.06% | 61.50% | 64.36% | 64.76% |
| Elim (delimiter: ##) | 62.50% | 65.90% | 67.96% | 69.46% |
| Elim (delimiter: \n\n) | 61.96% | 64.66% | 65.66% | 66.20% |

## D.2 ADDITIONAL EXPERIMENTAL RESULTS

### D.2.1 ADDITIONAL EXPERIMENTS ON MATH-500 AND LIVECODEBENCH

**Results of extended exploration rules on MATH-500.** Fig. 6 provides the coverage performance of UCB and GAP on the MATH-500 dataset; the `max_samples` hyperparameter for both UCB and GAP are provided in Table 2.

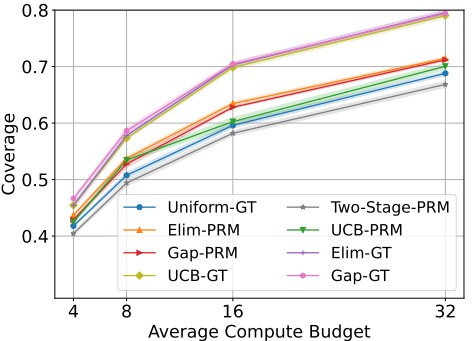 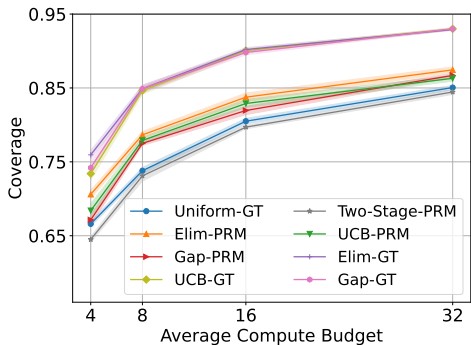

Figure 6: Results on MATH-500 with extended exploration rules. *Left:* Results with `Llama-3.2-1B-Instruct`. *Right:* Results with `Llama-3.1-8B-Instruct`.

Table 2: The choice of `max_samples` by model-scene and compute budget on MATH-500.

| Scene | Avg. budget 4 (max_samples) | Avg. budget 8 (max_samples) | Avg. budget 16 (max_samples) | Avg. budget 32 (max_samples) |
|---|---|---|---|---|
| `Llama-3.1` w/ GT | 40 | 40 | 120 | 300 |
| `Llama-3.1` w/ PRM | 12 | 40 | 80 | 120 |
| `Llama-3.2` w/ GT | 40 | 40 | 120 | 120 |
| `Llama-3.2` w/ PRM | 12 | 12 | 60 | 60 |

**Online compute allocation on MATH-500.** We further evaluate a streaming variant of Algorithm 1 in Table 3, where queries arrive sequentially. In this setting, compute is allocated on a per-query basis over 500 rounds, rather than jointly across the entire dataset. For the streaming variant, we tune the `max_samples` hyperparameter via a binary search over the grid $\{15 + 5k \mid k = 0, \dots, 37\}$. Even under the streaming constraint, our algorithm still outperforms uniform allocation with pool access.

Table 3: Results of the streaming variant of Algorithm 1 on MATH-500 with `Llama-3.2-1B-Instruct`.

| | Average compute budget | | | |
|---|---|---|---|---|
| **Method** | **4** | **8** | **16** | **32** |
| **Coverage** | | | | |
| Uniform-PRM | 41.80% | 50.75% | 59.60% | 68.80% |
| Elim-PRM | 43.70% | 53.75% | 63.45% | 71.45% |
| Elim-PRM (Streaming) | 42.65% | 52.85% | 62.95% | 71.05% |
| **Accuracy** | | | | |
| Uniform-PRM | 35.00% | 40.70% | 46.45% | 49.45% |
| Elim-PRM | 37.15% | 43.70% | 48.90% | 52.60% |
| Elim-PRM (Streaming) | 36.55% | 43.10% | 48.55% | 52.20% |

**Token-controlled evaluation.** To isolate the effect of allocation from differences in total generation length, we run a token-controlled variant of Algorithm 1. Specifically, we first record the total number of generated tokens under UNIFORM allocation, and then run Algorithm 1 while enforcing the same total token budget (token-matched). As shown in Table 4 (MATH-500) and Table 5 (LiveCodeBench), Algorithm 1 remains consistently better than under UNIFORM with the same number of tokens.

**Choices of the threshold $\gamma$.** The elimination threshold $\gamma$ controls which queries are confidently answered and can be removed from the active set. Since higher reward scores generally indicate higher-quality responses, setting a high threshold $\gamma \in [0, 1]$ is natural. We conduct ablations with $\gamma \in \{0.97, 0.98, 0.99, 1.0\}$ on MATH-500 and report the results in Fig. 7. We observe that $\gamma = 1.0$

Table 4: Token-matched evaluation on MATH-500 with `Llama-3.2-1B-Instruct`.

| Method | Average compute budget | | | |
|---|---|---|---|---|
| | **4** | **8** | **16** | **32** |
| **Coverage** | | | | |
| Uniform-PRM | 41.80% | 50.75% | 59.60% | 68.80% |
| Elim-PRM | 43.70% | 53.75% | 63.45% | 71.45% |
| Elim-PRM (Token Matched) | 42.70% | 52.10% | 60.90% | 69.95% |
| **Accuracy** | | | | |
| Uniform-PRM | 35.00% | 40.70% | 46.45% | 49.45% |
| Elim-PRM | 37.15% | 43.70% | 48.90% | 52.60% |
| Elim-PRM (Token Matched) | 36.50% | 42.60% | 48.05% | 51.85% |

Table 5: Token-matched evaluation on LiveCodeBench with `DeepSeek-R1-Distill-Llama-8B`.

| Method | Average compute budget | | | |
|---|---|---|---|---|
| | **4** | **8** | **16** | **32** |
| **Coverage** | | | | |
| Uniform-GT | 73.43% | 81.05% | 86.01% | 89.87% |
| Elim-GT | 84.66% | 91.02% | 94.00% | 94.00% |
| Elim-GT (Token Matched) | 84.50% | 91.65% | 94.00% | 94.00% |

performs slightly better, likely because `Qwen2.5-Math-PRM-7B` assigns a deterministic score of 1.0 to answers it deems correct—a property specific to this PRM. Importantly, across all tested values, our method consistently outperforms the uniform allocation baseline, indicating that Algorithm 1 is robust to variations in $\gamma$.

**Effect of PRM quality.** To study the impact of PRMs quality on the effectiveness of our method, we conducted additional experiments using a relatively weak PRM, `Qwen2.5-Math-7B-PRM800K`, in contrast to the strong PRM, `Qwen2.5-Math-PRM-7B`, employed in the main experiments. As shown in the Fig. 8, our method consistently outperforms the baselines across all compute budgets under both weak and strong reward models. Notably, performance improves with a stronger oracle, suggesting that our algorithm benefits directly from higher-quality reward signals.

### D.2.2 ADDITIONAL EXPERIMENTS AND ANALYSES ON MATH-500-HARD

Fig. 9 presents additional results on MATH-500-Hard datasets, including experiments with the GAP algorithm and experiments with the `Llama-3.1-8B-Instruct` model. These experiments show that our algorithms introduced in Section 3.3, particularly ENTROPY and UCB, achieve early advantages and outperform both UNIFORM and ELIMINATION.

**Why ENTROPY works well on challenging datasets?** We conduct a detailed analysis on understand why ENTROPY performs particularly well on MATH-500-Hard datasets. Upon inspecting model responses to challenging queries, we observe that unsolvable queries are more likely to yield invalid outputs (e.g., incomplete or improperly formatted), resulting in lower entropy among their generated responses; in contrast, solvable queries tend to generate more diverse outputs (see Table 6 and Table 7 for statistics computed from 64 responses per query). Since ENTROPY prioritizes queries with higher entropy, it naturally allocates more compute to those that are more likely to be solvable—explaining its strong empirical performance on challenging problems. We expect this behavior to generalize to other challenging benchmarks, provided that invalid responses can be reliably identified. In such settings, ENTROPY offers an effective means to shift compute toward promising queries and achieve better performance under limited compute budget.

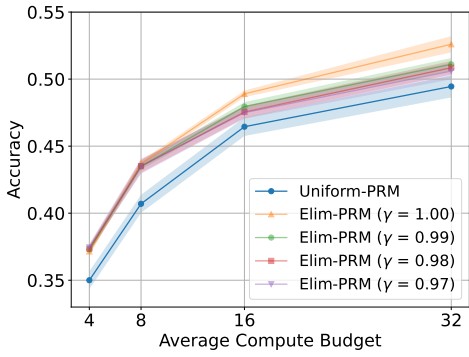

Figure 7: Ablation study for the hyperparameter $\gamma$. Experiments conducted on MATH-500 with `Llama-3.2-1B-Instruct`.

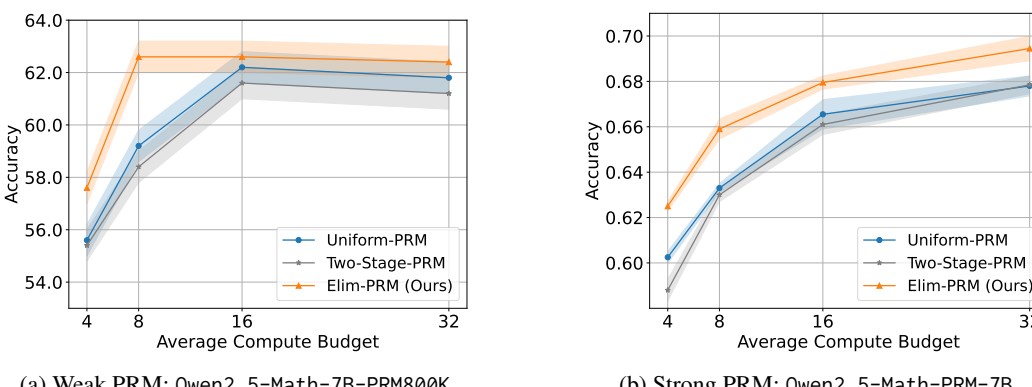

(a) Weak PRM: `Qwen2.5-Math-7B-PRM800K`

(b) Strong PRM: `Qwen2.5-Math-PRM-7B`

Figure 8: Effect of PRM quality on MATH-500 with `Llama-3.1-8B-Instruct`. We compare performance using a weaker PRM (`Qwen2.5-Math-7B-PRM800K`) versus a stronger PRM (`Qwen2.5-Math-PRM-7B`) across compute budgets.

Table 7: Aggregated statistics by query group on MATH-500-Hard-16.

| Query group | #questions | Entropy (mean) | Invalid answers (%) |
|---|---|---|---|
| Unsolvable | 43 | 4.33 | 18.39% |
| Solvable | 13 | 4.53 | 13.10% |

**Hyperparameter study of $\lambda$ for ENTROPY and UCB.** We study the effect of the exploration coefficient $\lambda$ in our Entropy- and UCB-based extension rules on the MATH-500-Hard-8 dataset. Results in Tables 8 and 9 suggest that ENTROPY benefits from larger $\lambda$, while UCB prefers smaller $\lambda$.

Table 8: Effect of $\lambda$ on Entropy-based Allocation.

| $\lambda$ choice / Avg compute budget | 4 | 8 | 16 | 32 |
|---|---|---|---|---|
| Entropy ($\lambda = 1.0$) | 0.0% | 5.4% | 8.9% | 17.9% |
| Entropy ($\lambda = 2.0$) | 0.0% | 5.4% | 8.9% | 17.9% |
| Entropy ($\lambda = 3.0$) | 0.0% | 5.4% | 8.9% | 21.4% |

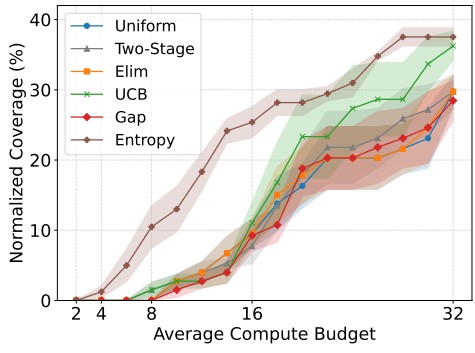

(a) Coverage with `Llama-3.2-1B-Instruct` on MATH-500-Hard-8

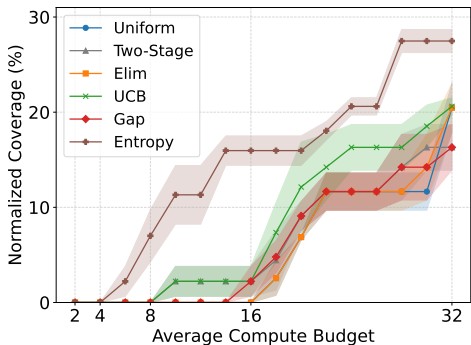

(b) Coverage with `Llama-3.2-1B-Instruct` on MATH-500-Hard-16

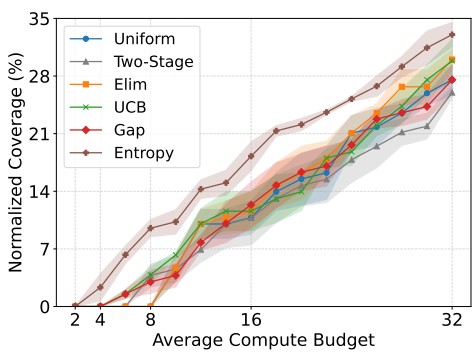

(c) Coverage with `Llama-3.1-8B-Instruct` on MATH-500-Hard-8

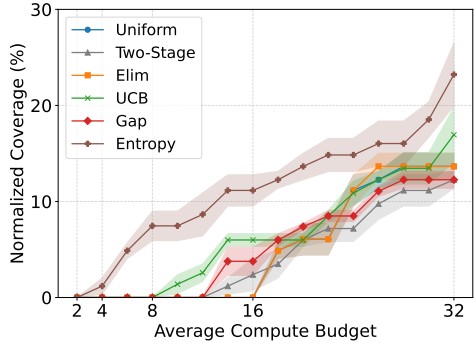

(d) Coverage with `Llama-3.1-8B-Instruct` on MATH-500-Hard-16

Figure 9: Coverage comparisons on MATH-500-Hard datasets with two language model of different sizes: `Llama-3.2-1B-Instruct` and `Llama-3.1-8B-Instruct`.

Table 6: Aggregated statistics by query group on MATH-500-Hard-8.

| Query group | #questions | Entropy (mean) | Invalid answers (%) |
|---|---|---|---|
| Unsolvable | 49 | 4.26 | 19.45% |
| Solvable | 22 | 4.52 | 12.45% |

Table 9: Effect of $\lambda$ on UCB-based Allocation.

| $\lambda$ choice / Avg compute budget | 4 | 8 | 16 | 32 |
|---|---|---|---|---|
| UCB ($\lambda = 1.0$) | 0.0% | 1.8% | 3.6% | 16.1% |
| UCB ($\lambda = 2.0$) | 0.0% | 1.8% | 3.6% | 14.3% |
| UCB ($\lambda = 3.0$) | 0.0% | 1.8% | 1.8% | 12.5% |

# E   THE USE OF LARGE LANGUAGE MODELS (LLMs)

LLMs were used to polish the writing of this paper.

