# OpenReview forum: "Strategic Scaling of Test-Time Compute: A Bandit Learning Approach"
_ICLR.cc/2026/Conference — ICLR 2026 Poster_

### Official Review · Reviewer_7ofK · 2025-10-31

**Soundness:** 3
**Presentation:** 3
**Contribution:** 2
**Rating:** 6
**Confidence:** 3

**Summary:**

This paper addresses the inefficiency of uniform test-time compute allocation for LLMs and proposes a bandit learning framework for strategic compute distribution. It formulates test-time compute allocation as a bandit problem, treating each query as an action and allocating compute sequentially to maximize the number of correctly answered queries. Theoretical analysis and experiments validates the algorithm’s superior compute efficiency over uniform allocation.

**Strengths:**

1) The paper formulates test-time compute allocation as a bandit problem and demonstrates its superiority over uniform allocation through both theoretical and empirical analyses.
2) The framework supports multiple exploration strategies, aggregation methods, and settings, ensuring broad applicability.

**Weaknesses:**

1) The computational overhead of calculating Entropy and UCB should also be taken into account; comparing only the inference budget is insufficient. Explicit end-to-end run-time comparisons with each baseline are necessary.
2) The impact of the Entropy and UCB hyper-parameter $\lambda$ has not been thoroughly analyzed; ablation experiments showing how $\lambda$ affects performance across different tasks should be provided.

**Questions:**

Please refer to the "Weakness" part.

---

> ### Author Response · Authors · 2025-11-25
> **Author response**
>
> Dear Reviewer,
>
> Thank you for taking the time to review our paper. Below, we provide detailed responses and clarifications. If our responses address your concerns, we would appreciate it if you could consider updating your assessment.
>
> **Response to Weaknesses:**
>
> >**Weakness 1:** The computational overhead of calculating Entropy and UCB should also be taken into account; comparing only the inference budget is insufficient. Explicit end-to-end run-time comparisons with each baseline are necessary.
>
> **Responses:** Thank you for your suggestion. We’d like to clarify that **the computational overhead of computing Entropy and UCB is negligible compared to the cost of LLM inference**. To verify this, we measured the actual runtime of both scoring computations and compared them directly with LLM inference time. Using the Llama3.2-1B-Instruct model, we ran 100 independent repetitions for both the scoring computations and the model inferences to obtain reliable average timing estimates:
>
>   - **UCB score computation time:** 2.28 seconds
>   - **Entropy score computation time:** 3.58 seconds
>   - **LLM inference time:** 857.40 seconds
>
> These results show that the overhead from UCB and Entropy scoring is **less than 0.42%** of the total runtime. Since LLM inference overwhelmingly dominates compute cost, these scoring computations do not meaningfully affect end-to-end runtime. We will clarify this point in our revised manuscript and include the timing analysis for completeness.
>
> >**Weakness 2:** The impact of the Entropy and UCB hyper-parameter $\lambda$  has not been thoroughly analyzed; ablation experiments showing how affects performance across different tasks should be provided.
>
> **Responses:** Thank you for raising this point. The hyperparameter $\lambda$ controls the balance between the two terms in UCB and Entropy: a larger $\lambda$ encourages more exploration, while a smaller $\lambda$ emphasizes exploitation (score-based exploitation for UCB and entropy-based exploitation for Entropy). In practice, we find that choosing $\lambda$ as a small constant works well—which is also consistent with standard practice in the bandit literature. The exact values used in our experiments are reported in Appendix D.1.1.
>
> To address your concern more directly, we additionally conducted an ablation study on the MATH-500-Hard dataset using a range of $\lambda$ values. The results show that Entropy tends to benefit from moderately larger $\lambda$, while UCB performs best with smaller $\lambda$.
>
> **Effect of $\lambda$ on Entropy:**
> | **$\lambda$ choice / Avg compute budget** | **4**     | **8**     | **16**    | **32**    |
> |----------------------------------|-----------|-----------|-----------|-----------|
> | **Entropy ($\lambda$ = 1.0)**            | 0.0%      | 5.4%      | 8.9%      | 17.9%     |
> | **Entropy ($\lambda$ = 2.0)**            | 0.0%      | 5.4%      | 8.9%      | 17.9%     |
> | **Entropy ($\lambda$ = 3.0)**            | 0.0%      | 5.4%      | 8.9%      | 21.4%     |
>
> **Effect of $\lambda$ on UCB:**
> | **$\lambda$ choice / Avg compute budget** | **4**     | **8**     | **16**    | **32**    |
> |----------------------------------|-----------|-----------|-----------|-----------|
> | **UCB ($\lambda$ = 1.0)**                | 0.0%      | 1.8%      | 3.6%      | 16.1%     |
> | **UCB ($\lambda$ = 2.0)**                | 0.0%      | 1.8%      | 3.6%      | 14.3%     |
> | **UCB ($\lambda$ = 3.0)**                | 0.0%      | 1.8%      | 1.8%      | 12.5%     |

---

### Official Review · Reviewer_q1r5 · 2025-11-01

**Soundness:** 2
**Presentation:** 2
**Contribution:** 2
**Rating:** 2
**Confidence:** 4

**Summary:**

This paper tackles the problem of test-time compute allocation for large language models (LLMs). Traditional approaches such as Best-of-N sampling uniformly allocate the same amount of compute to each query, regardless of difficulty. The authors argue that this is inefficient, as some queries are trivially solvable while others are much harder. They reformulate the problem as a bandit learning task, where each query is an “arm” and compute allocation is treated as exploration. Their proposed algorithms estimate query difficulty adaptively and allocate compute dynamically across queries. The core method progressively eliminates solved queries, redistributing compute to unsolved or more uncertain ones. Extensions include exploration variants, support for streaming queries, and token-level compute control. The authors provide theoretical results showing superior compute efficiency over uniform allocation and validate the method empirically on math and code benchmarks, achieving improvements under the same compute budget.

**Strengths:**

- The idea of using bandit on allocating compute budget seems relatively novel.
- A theory is provided.

**Weaknesses:**

- The major concern is that the authors should use stronger PRMs as reward "oracles", e.g. Reasonflux-PRM [1]. On the one hand, Qwen2.5-Math-PRM-7B is relatively old compared with the latest SOTA PRMs. On the other hand, Qwen3-1.7B and DeepSeek-R1-Distill-Llama-8B are reasoning models, thus using Qwen2.5-Math-PRM-7B, which is not trained for reasoning models, as the reward oracle may lead to suboptimal results. Reasonflux-PRM is a more recent SOTA PRM that can be used as the reward oracle to further validate the effectiveness of the proposed method on both non-reasoning and reasoning models.
- Another concern on experiments is that self-consistency [2] is also a widely used algorithm to aggregate multiple response without PRMs, it will also be helpful to see how the proposed method will perform with that, such as [3].
- The current paper fails to provide a systematic related work. While it is understandable that test-time scaling algorithms are a relatively new concept and most work can be classified as concurrent and does not need to be compared, it is not acceptable not to formally mention them.


[1] ReasonFlux-PRM: Trajectory-Aware PRMs for Long Chain-of-Thought Reasoning in LLMs.

[2] Self-Consistency Improves Chain of Thought Reasoning in Language Models.

[3] Scaling LLM Test-Time Compute Optimally Can be More Effective than Scaling Parameters for Reasoning.

**Questions:**

Please refer to the weaknesses above.

---

> ### Author Response · Authors · 2025-11-25
> **Author response**
>
> Dear Reviewer,
>
> Thank you for taking the time to review our paper. Below, we provide detailed responses and clarifications. If our responses address your concerns, we would appreciate it if you could consider updating your assessment.
>
> **Response to Weaknesses:**
>
> >**Weakness 1:** The major concern is that the authors should use stronger PRMs as reward "oracles", e.g. Reasonflux-PRM [1]. On the one hand, Qwen2.5-Math-PRM-7B is relatively old compared with the latest SOTA PRMs. On the other hand, Qwen3-1.7B and DeepSeek-R1-Distill-Llama-8B are reasoning models, thus using Qwen2.5-Math-PRM-7B, which is not trained for reasoning models, as the reward oracle may lead to suboptimal results. Reasonflux-PRM is a more recent SOTA PRM that can be used as the reward oracle to further validate the effectiveness of the proposed method on both non-reasoning and reasoning models.
>
> **Responses:** Thank you for your comment. We would like to clarify that we already include the **ground-truth (GT) oracle**—the strongest possible reward oracle—in our experiments for both reasoning and non-reasoning models. Under this strongest reward oracle, our algorithm consistently outperforms all baselines, further validating its effectiveness.
>
> >**Weakness 2:** Another concern on experiments is that self-consistency [2] is also a widely used algorithm to aggregate multiple response without PRMs, it will also be helpful to see how the proposed method will perform with that, such as [3].
>
> **Responses:** Thank you for your comment. We have already included **self-consistency (SC)** as an alternative aggregation method in our algorithm. This extension is described in Section 3.3 (lines 237–245), with corresponding results in Figure 1 (middle) and Figure 3 (left). Under SC, our adaptive algorithm still significantly outperforms SC-based baselines. The two papers mentioned in the review are also discussed in our main text and included in our references.
>
> >**Weakness 3:** The current paper fails to provide a systematic related work. While it is understandable that test-time scaling algorithms are a relatively new concept and most work can be classified as concurrent and does not need to be compared, it is not acceptable not to formally mention them.
>
> **Responses:** Thank you for your comment. As stated in the introduction (lines 89–90), we present the related work section in the Appendix due to space constraints.

---

### Official Review · Reviewer_pUPq · 2025-11-02

**Soundness:** 3
**Presentation:** 4
**Contribution:** 2
**Rating:** 4
**Confidence:** 3

**Summary:**

This paper studies online algorithms for optimizing test-time compute allocation across a set of queries. Instead of uniformly allocating compute, the authors formulate the problem as a bandit learning task and develop an adaptive framework that learns to prioritize solvable instances, thereby avoiding excessive computation on unsolvable queries. The paper provides theoretical analysis showing improved compute efficiency over uniform allocation and demonstrates empirical gains on math and coding datasets.

**Strengths:**

1. The paper is clearly written and easy to follow.
2. The proposed algorithmic framework is simple yet effective. It is modular and flexible—different components such as exploration and exploitation strategies can be adjusted to fit various scenarios.
3. The work presents both theoretical and empirical results, and the experiments show consistent improvements over uniform allocation baselines.

**Weaknesses:**

1. While simplicity is a strength, the algorithmic contribution feels somewhat incremental. The framework extends a single-query setting to multiple queries in a relatively straightforward manner, and the theoretical analysis relies on standard probabilistic arguments. The results mainly show that the proposed method outperforms uniform allocation—which is expected—but do not clarify whether it outperforms other adaptive baselines such as ETC from a theoretical standpoint, or whether the approach is optimal or near-optimal.
2. The assumption of having access to an accurate reward model and threshold parameter $\gamma$ seems restrictive. It would be valuable to discuss how the method behaves or could be adapted under model inaccuracies.
3. The current analysis focuses on a high-budget regime where every query can, in principle, be solved correctly. In realistic scenarios with a limited budget $B$, it would be more informative to characterize how many queries are expected to be solved correctly given small $B$, or how the algorithm trades off between query coverage and accuracy.

**Questions:**

Please refer to the above weakness part.

---

> ### Author Response · Authors · 2025-11-25
> **Author response (1/2)**
>
> Dear Reviewer,
>
> Thank you for taking the time to review our paper. Below, we provide detailed responses and clarifications. If our responses address your concerns, we would appreciate it if you could consider updating your assessment.
>
> **Response to Weaknesses:**
>
> >**Weakness 1:** While simplicity is a strength, the algorithmic contribution feels somewhat incremental. The framework extends a single-query setting to multiple queries in a relatively straightforward manner, and the theoretical analysis relies on standard probabilistic arguments. The results mainly show that the proposed method outperforms uniform allocation—which is expected—but do not clarify whether it outperforms other adaptive baselines such as ETC from a theoretical standpoint, or whether the approach is optimal or near-optimal.
>
> **Responses:** Thank you for your comment. We clarify below both our main contributions and the theoretical guarantees, including a comparison against ETC-style baselines.
>
> First, our main contributions are (i) **adapting fully adaptive bandit algorithms to test-time compute allocation** and (ii) **showing that this leads to substantial gains over existing strategies**, particularly uniform allocation and two-stage heuristics such as ETC. To our knowledge, this adaptation is new and opens the door to further exploration of bandit-based adaptive test-time scaling. In addition, our work introduces a **novel bandit formulation tailored to test-time compute allocation**, where the objective is to generate at least one correct response for each query—distinct from classical pure exploration bandit objectives, which aim to identify high-reward arms. Please see lines 187–194 for more details.
>
> Second, our budget complexity $\tilde O\left(\sum_x \tfrac{1}{\Delta_x}\right)$ is **near-optimal**, up to logarithmic factors. Appendix C shows that if an algorithm spends less than $\tfrac{1}{\Delta_x}$ compute on some query $x$, then with constant probability it never observes a correct generation for $x$, and hence outputs an incorrect response for that query (see lines 966–980). Therefore, any method that aims to correctly answer all queries with high probability must spend at least $\Omega(\tfrac{1}{\Delta_x})$ compute on each $x$, for a total budget of $\Omega\left(\sum_x \tfrac{1}{\Delta_x}\right)$. This implies that our budget complexity is near-optimal; we will clarify this in the revision.
>
> Third, **the two-stage explore-then-commit (ETC) algorithm is theoretically sub-optimal** as it performs *uniform budget allocation* during its first stage (i.e., the exploration stage). Suppose ETC allocates a fixed exploration budget $m$ to every query in the first stage. Let $n_x$ denote the total number of generations for query $x$ across both stages. Because the first stage already draws $m$ samples for each $x$, we have $n_x \geq m$ for all $x$. Meanwhile, by the same per-query lower bound above, correctly answering $x$ with high probability requires $n_x \ge \Omega(\tfrac{1}{\Delta_x})$. Therefore, any ETC-style algorithm must incur a total budget of $\Omega (\sum_{x} \max(m, \tfrac{1}{\Delta_x}))$, which is strictly larger than our budget complexity whenever $m$ is nontrivial and many queries are easy (i.e., when $\tfrac{1}{\Delta_x}$ is small). In contrast, our algorithm is **fully adaptive** and avoids this inefficiency. We will add a discussion of this point in the revision.
>
> Finally, **we already include an ETC-style baseline in our experiments (denoted “Two-Stage”)**. As shown in Figure 1 (left) and Figure 2, this ETC baseline is substantially outperformed by our fully adaptive algorithm, consistent with the theoretical analysis above.

---

> > ### Author Response · Authors · 2025-11-25
> > **Author response (2/2)**
> >
> > >**Weakness 2:** The assumption of having access to an accurate reward model and threshold parameter $\gamma$ seems restrictive. It would be valuable to discuss how the method behaves or could be adapted under model inaccuracies.
> >
> > **Responses:**  Thank you for your comment. We first note that *relying on a reward model is not unique to our method*: even standard uniform Best-of-N uses a reward model to choose the final answer. If the reward model is weak or inaccurate, both our method and uniform allocation will be affected. The key message from our experiments is that, *under the same reward model*, our fully adaptive algorithm consistently outperforms baselines.
> >
> > To further address this concern, we conducted additional experiments using a **weaker PRM** (Qwen2.5-Math-7B-PRM800K) in contrast to the **strong PRM** (Qwen2.5-Math-PRM-7B) used in the main experiments. As shown in the tables below, our method continues to outperform both Uniform and Two-Stage baselines across all compute budgets and **under both weak and strong reward models**.
> >
> > We also highlight that our paper already discusses an extension of Algorithm 1 that replaces the reward model with **self-consistency (SC)** as the aggregation rule (see lines 237–245). This version removes the need for a reward model entirely. Empirically (as in Figure 1 middle plot and Figure 3 left plot), our adaptive algorithm still significantly outperforms SC-based baselines, further demonstrating the robustness of our approach.
> >
> > Finally, the threshold parameter $\gamma$ is treated as a hyperparameter. We include an ablation study in the appendix (Figure 7), which shows that for various $\gamma$ choices, our method consistently outperforms the baselines.
> >
> > **Performance comparison on MATH-500 under the weak PRM**
> > | **Method / Avg compute budget** | **4**       | **8**       | **16**      | **32**      |
> > |-----------------------------|---------|---------|---------|---------|
> > | **Weak PRM (Uniform)**             | 55.6%   | 59.2%   | 62.2%   | 61.8%   |
> > | **Weak PRM (Two-Stage)**             | 55.4%   | 58.4%   | 61.6%   | 61.2%   |
> > | **Weak PRM (Ours)**             | **57.6%**   | **62.6%**   | **62.6%**   | **62.4%**   |
> >
> > **Performance comparison on MATH-500 under the strong PRM**
> > | **Method / Avg compute budget** | **4**       | **8**       | **16**      | **32**      |
> > |-----------------------------|---------|---------|---------|---------|
> > |  **Strong PRM (Uniform)**            | 60.0%   | 63.6%   | 67.2%   | 68.4%   |
> > |  **Strong PRM (Two-Stage)**            | 58.2%   | 62.4%   | 66.8%   | 67.8%   |
> > |  **Strong PRM (Ours)**       | **62.0%**   | **66.4%**   | **68.2%**   | **70.2%**   |
> >
> >
> > >**Weakness 3:** The current analysis focuses on a high-budget regime where every query can, in principle, be solved correctly. In realistic scenarios with a limited budget $B$, it would be more informative to characterize how many queries are expected to be solved correctly given small $B$, or how the algorithm trades off between query coverage and accuracy.
> >
> > **Responses:** Thank you for your insightful comment. Our theoretical analysis focuses on quantifying the budget complexity required to correctly answer all queries, with the goal of providing clean insights into the compute efficiency of different algorithms. A full theoretical treatment of the low-budget setting is an interesting direction for future work. That said, we did empirically analyze the behavior of our algorithms under limited budgets in Section 4.3 (lines 411–431). For standard datasets that include both easy and hard (but solvable) queries, our algorithms allocate more compute to challenging queries while maintaining accuracy on easier ones. For challenging datasets that include both solvable and unsolvable queries, our algorithms learn to prioritize solvable instances, effectively reducing excessive compute spent on unsolvable queries. In both settings, our algorithms achieve better overall performance than the baselines.

---

### Meta-Review · Area_Chair_XB9a · 2026-01-09

**Summary:**

This paper formulates the test-time compute allocation across a set of queries as a bandit learning task and proposes an adaptive framework that learns to prioritize solvable instances, which avoids excessive computation on unsolvable queries. The paper provides a theoretical analysis showing improved computational efficiency over uniform allocation and demonstrates empirical gains on math and coding datasets.

The reviewers think that the proposed method is simple but novel, with supported theoretical analysis and empirical validation. The main concerns include the assumption of access to accurate reward models and high budgets, and a lack of analysis on computational overhead and hyper-parameter studies. The responses from the authors have clarified these concerns, and I believe that the work is a good contribution.

**Reviewer Concerns:**

- pUPq
   - the algorithmic contribution feels somewhat incremental.
   - assumption of having access to an accurate reward model and threshold parameter $\lambda$ seems restrictive
   - current analysis focuses on a high-budget regime, where every query can potentially be solved correctly.
- q1r5
   - should use stronger PRMs as reward "oracles" (Qwen2.5-Math-PRM-7B is relatively old). Reasonflux-PRM is a more recent SOTA PRM
   - Another concern on experiments is that self-consistency [2] is also a widely used algorithm to aggregate multiple response without PRMs
   - fails to provide a systematic related work
- 7ofK
   - The computational overhead of calculating Entropy and UCB should also be taken into account
   -  impact of the Entropy and UCB hyper-parameter  has not been thoroughly analyzed

**Reviewer Scores:**

The comments have been addressed, so I believe the reviewers will either keep the (positive) score or increase the original below-acceptance score. Nevertheless, the reviewer pUPq also thinks that the novelty of the work is limited.
    - pUPq: 4
    - q1r5: 2
    - 7ofK: 6

---

### Decision · Program_Chairs · 2026-01-26

Accept (Poster)